# Preconception Counseling in Patients with Hypothyroidism and/or Thyroid Autoimmunity

**DOI:** 10.3390/medicina58081122

**Published:** 2022-08-18

**Authors:** Mihaela Țarnă, Luminița Nicoleta Cima, Anca Maria Panaitescu, Carmen Sorina Martin, Anca Elena Sîrbu, Carmen Gabriela Barbu, Bogdan Pavel, Andreea Nicoleta Șerbănică, Simona Fica

**Affiliations:** 1Department of Endocrinology and Diabetes, Nutrition and Metabolic Diseases—“Elias” Emergency University Hospital, 011461 Bucharest, Romania; 2Department of Endocrinology, Carol Davila University of Medicine and Pharmacy, 050474 Bucharest, Romania; 3Department of Obstetrics and Gynecology Filantropia Clinical Hospital, 011171 Bucharest, Romania; 4Department of Obstetrics and Gynecology, Carol Davila University of Medicine and Pharmacy, 050474 Bucharest, Romania; 5Department of Functional Sciences, Carol Davila University of Medicine and Pharmacy, 050474 Bucharest, Romania; 6Department of Pediatrics, Carol Davila University of Medicine and Pharmacy, 050474 Bucharest, Romania; 7Department of Pediatrics Hematology, Fundeni Clinical Institute, 022328 Bucharest, Romania

**Keywords:** hypothyroidism, subclinical hypothyroidism (SCH), thyroid autoimmunity, iodine status, screening, preconception, pregnancy

## Abstract

Preconception counseling is an essential tool for preventing adverse pregnancy outcomes associated with thyroid dysfunction. The high prevalence of thyroid disease among women of reproductive age, and the increased risk of adverse pregnancy outcomes associated with thyroid dysfunction, emphasize the necessity for well-established screening and treatment criteria in the preconception period. We therefore conducted a literature review for relevant information on the screening, diagnosis and treatment of subclinical and overt hypothyroidism in women seeking pregnancy. While screening for thyroid disease is recommended only in the presence of risk factors, iodine supplementation should be recommended in most regions, with higher doses in areas with severe deficiency. Known hypothyroid women should be counseled about increasing their levothyroxine dose by 20–30% in the case of suspected or confirmed pregnancy (missed menstrual cycle or positive pregnancy test). Treating subclinical hypothyroidism appears to be beneficial, especially in the presence of autoimmunity or in patients undergoing artificial reproductive techniques. Regarding the management of TPOAb negative SCH women or euthyroid women with positive TPOAb, further research is necessary in order to make evidence-based recommendations.

## 1. Introduction

Thyroid disease is the most common endocrine disorder that affects women of reproductive age. Suboptimal thyroid function in pregnancy carries a significant risk of adverse pregnancy outcomes, including miscarriages, stillbirths and neuro-intellectual impairment of the offspring. These events occur mostly in patients with overt thyroid disease, but also in patients with borderline abnormalities, such as subclinical hypothyroidism or thyroid autoimmunity [1]. Preconception evaluation of reproductive age woman with thyroid disease should be made with regards to the physiologic adaptations that appear in pregnancy, as well as to the importance of thyroid hormones in fetal development. Our goal is to raise awareness among health care physicians about the importance of preconceptional counseling, and consequently to improve the prevention of pregnancy complications and adverse fetal outcomes related to thyroid dysfunction.

## 2. Methods

We conducted a literature review of medical databases, including PubMed, Google Scholar and Cochrane Controlled trials Register for relevant information on the screening, diagnosis and treatment of subclinical and overt hypothyroidism during preconception period, and the impact they have on fertility, pregnancy and newborn. The key words used for the search included: hypothyroidism, subclinical hypothyroidism (SCH), thyroid autoimmunity, iodine status, screening, preconception, pregnancy. We prioritized the latest guidelines, most cited reviews and meta-analyses, and most recent randomized controlled trials (RCT).

## 3. Screening for Thyroid Disease

The issue of universal thyroid screening before or at the beginning of pregnancy is still a matter of debate since insufficient data are currently available. High prevalence of the condition, adverse health outcomes and effective treatment, along with cost-efficient screening, are the criteria that must be fulfilled in order to recommend a general screening. Thyroid disease partially accomplishes these criteria, although further research is required in order to make evidence-based recommendations.

The most frequently used blood tests for the assessment of thyroid status include thyroid stimulating hormone (TSH), free T4 (FT4), total T4 (TT4) and thyroid peroxidase antibodies (TPOAb), which are relatively inexpensive and widely available. Using appropriate population-based reference ranges, where available, would be preferable [2]. Moreover, thyroid disease, mainly hypothyroidism and thyroid autoimmunity are common conditions, with prevalence rates of 2–3% and up to 17% in reproductive age woman, respectively [2,3]. Untreated overt hypothyroidism has been associated with pregnancy risks and adverse fetal outcomes, such as pregnancy loss, stillbirth and impaired neurocognitive development [4]. However, much more prevalent than overt thyroid disease is subclinical hypothyroidism, with a large proportion of patients that will be diagnosed in the case of a universal screening, raising the question about the necessity and effectiveness of substitutive treatment in such case. Screening all women for thyroid disease or autoimmunity preconception may contribute to overdiagnosis of hypothyroidism and overtreatment during pregnancy and post-partum [5,6].

Although universal screening may prove beneficial, existing data are not sufficient to support this testing algorithm at present. Therefore, American Thyroid Association (ATA) guidelines recommend verbally screening and clinical evaluation of all patients seeking pregnancy or newly pregnant, in order to identify women who associate any of the following risk factors (Figure 1) for thyroid disease [2,7]:A history of thyroid dysfunction such as hypothyroidism/hyperthyroidism, known thyroid antibody positivity or prior thyroid surgery;Current symptoms or signs of thyroid dysfunction or presence of a goiter;History of head or neck radiation, including radioactive iodine treatment;Use of amiodarone or lithium, or recent administration of iodinated radiologic contrast;Residing in an area of known moderate-to-severe iodine insufficiency;Family history of autoimmune thyroid disease or thyroid dysfunction;Age >30 years (The prevalence of hypothyroidism increases with age);Multiple prior pregnancies (≥2);History of pregnancy loss, preterm delivery, or infertility;Morbid obesity (BMI ≥40 kg/m^2^);Type 1 diabetes or other autoimmune disorders associated with autoimmune thyroid dysfunction (vitiligo, adrenal insufficiency, hypoparathyroidism, atrophic gastritis, pernicious anemia, systemic sclerosis, systemic lupus erythematosus, and Sjögren’s syndrome).

If any of the above-mentioned risk factors are identified, testing for serum TSH is recommended, and if TSH is elevated (2.5–10 mU/L), then FT4 and TPOAb or ultimately thyroglobulin antibodies (TgAb) should be determined. Thyroid ultrasound could help evaluating characteristic sonographic features [2,8].

## 4. Iodine Status

### 4.1. Epidemiology

Iodine deficiency is the most common cause of preventable brain damage and cognitive impairment in children. It is estimated that over 1.8 billion people worldwide are at risk of iodine deficiency [9]. The World Summit for Children (New York, NY, USA, 1990) promoted Universal salt iodization as the main strategy for eliminating iodine deficiency, which conducted to an important progress for reducing iodine deficiency prevalence [8,10].

In 2013, inadequate iodine supply in pregnancy was present in up to 30% of the European countries (e.g., Albania, Belgium, Czech Republic, Greece, Israel, Norway, Portugal, Romania, Serbia, France, Hungary, Ireland, Italy and the UK) [11]. Legislation regarding salt iodination has led to significant improvements in iodine status. The number of countries worldwide with adequate iodine intake has nearly doubled from 67 in 2003 to 118 in 2020 [12]. However, pregnant or breastfeeding women need extra iodine, which puts them at greater risk of deficiency. They have an increased thyroid hormone production that has to cover both mother and fetal needs, fetal iodine requirements for its own thyroid hormone synthesis, increased renal iodine excretion, and the breast milk secretion of iodine for the infant [9]. For example, in Romania, mild iodine deficiency was also still prevalent among pregnant women from historical endemic regions, mainly in the Carpathian and sub-Carpathian areas, after 13 years of universal salt iodization. Therefore, it is important to know the actual iodine status of each region and encourage iodine supplementation during pregnancy in deficient areas [13,14,15].

### 4.2. Effects on Pregnancy and Fetal/Newborn Outcomes

Iodine is essential for thyroid hormone synthesis, and even if needed in small amounts, its deficiency before or during pregnancy results in impaired maternal and fetal thyroid hormone synthesis [16]. This is associated with maternal and fetal goiter due to increased TSH stimulation [17], increased rates of pregnancy loss, stillbirth, and increased perinatal and infant mortality in cases of severe deficiency, and also cognitive impairment of the offspring [18,19,20,21]. Cretinism is the most severe form of cognitive impairment, and it is characterized by profound intellectual deficiencies, deaf-mutism, and motor rigidity. Fortunately, since the implementation of universal salt iodization, the incidence of cretinism has been considerably reduced [22].

### 4.3. Who and How to Treat?

Women with adequate iodine intake before and during pregnancy will have a stable level of iodine during pregnancy, with adequate deposits of intrathyroidal iodine and no difficulty adapting to the increased demand for thyroid hormone throughout pregnancy. However, even in mild to moderate iodine-deficient areas, total-body iodine stores in pregnant women, as reflected by urinary iodine values between 50–150 µg/L, declines gradually during the pregnancy. Accordingly, in areas of iodine deficiency, iodine supplementation of mothers prior to conception or in early pregnancy should be recommended in order to improve children’s cognitive performance [23,24,25,26].

Iodine sources come from diet and vitamin/mineral preparations. In most regions, women who are planning pregnancy or currently pregnant, should supplement their diet with a daily oral supplement of potassium iodide that contains 150 µg of iodine, or even 250 µg in areas with severe iodine deficiency [2,27].

#### Thyroid Disease

The main cause of hypothyroidism in iodine-replete countries is Hashimoto’s thyroiditis, while iodine deficiency remains relevant in areas with severe deficiency. Other common causes include thyroidectomy and radioiodine therapy for Graves’ disease or for benign and malignant thyroid disease [1].

Hashimoto disease is the most frequent autoimmune thyroid disease, characterized by the presence of cell and humoral immune response against thyroid antigens. It involves infiltration of thyroid tissue with T cells and B cells, autoantibodies production, including thyroid TPOAb and TgAb, which finally lead to impaired hormone production and different clinical manifestations [28].

## 5. Overt Hypothyroidism

### 5.1. Epidemiology/Definition/Diagnostic

Overt hypothyroidism, characterized by low FT4 and elevated TSH levels, affects 0.3–0.5% of the population and is ten times more prevalent in females than in males [1], while up to 2–3% of healthy, nonpregnant women of childbearing age have an elevated serum TSH. The most frequent causes are autoimmune thyroiditis and iodine deficiency. There may also be other rare causes, such as thyroidectomy for thyroid nodules or Graves’ disease [2,29,30].

### 5.2. Effects on Fertility, Pregnancy and Fetal/Newborn Outcomes

Optimal thyroid function is essential for a successful conception and pregnancy. In hypothyroidism, most evidence appears to support an association with an increased risk of infertility. Hypothyroid women with TSH concentrations >15 mU/L have a significant higher rate of irregular menses of 68%, compared with a 12% rate of menstrual irregularities reported by euthyroid women [31].

Undoubtedly, uncorrected overt hypothyroidism increases the risk of adverse pregnancy, fetal and neonatal outcomes such as eclampsia, anemia, placental abruption, postpartum hemorrhage, intrauterine fetal death, miscarriages, premature births, low birth weight and increased neonatal respiratory distress. Maternal hypothyroidism also affects neuro-intellectual and behavioral development of the child [1,32]. Moreover, TPOAb positive women appear to be at greater risk for adverse events compared to those who are TPOAb negative, even when thyroid function is identical [2,33].

The fetal thyroid gland starts thyroid hormone synthesis after the age of 12 weeks and is not fully functional until 18 to 20 weeks of gestation. Thus, for the first half of pregnancy, the fetus is dependent on the maternal supply of thyroxine [34].

### 5.3. Who and How to Treat?

Untreated hypothyroid women seeking pregnancy should immediately undergo substitutive treatment, in order to improve fertility and prevent adverse pregnancy and fetal outcomes.

Hypothyroid women treated with LT4 who are planning pregnancy, should have a serum TSH performed preconception, and LT4 dose adjusted to achieve a TSH value between the lower reference limit and 2.5 mU/L. Lower preconception TSH values (<1.5 mU/L) could reduce the risk of TSH elevation during the first trimester, but they are not recommended due to potential unknown side effects [2].

From a practical perspective, half of pregnancies in the general population are unplanned and more than that, pregnancy may not be recognized until mid-gestation. Moreover, the average initial antenatal evaluation is often delayed, at about 14 weeks of gestation [35]. For this reason, every hypothyroid woman of reproductive age, whether or not they are planning immediate pregnancy, should be educated regarding the increased demand for LT4 in case of pregnancy. They should be informed that in case of suspected or confirmed pregnancy (a missed menstrual cycle or positive home pregnancy test), they should independently increase the LT4 dose by 20%–30% (e.g., doubling dose on 2 days a week corresponds to approximately 30% increase [36] and contact their caregiver immediately for prompt testing and further evaluation [2].

Special attention should be offered to patients with thyroid nodules. Although pregnancy seems to have a permissive or stimulative effect on thyroid nodules, the prognosis of differentiated thyroid cancer in patients diagnosed during pregnancy or those who got pregnant after curative treatment, seems to be unaffected by pregnancy. Differentiated thyroid cancer should not discourage intended pregnancy nor indicate abortion. Conception should, however, occur after remission is confirmed and after adequate thyroid hormone replacement has been attained [37,38]. In the case of radioiodine treatment, this may lead to suboptimal thyroid hormonal control during the months following administration, so a minimum of 6 months to ensure that thyroid hormonal control is stable before conceiving should be envisioned [2]. The monitoring of a patient during pregnancy who has already been treated for differentiated thyroid cancer prior to pregnancy should be performed in accordance with the current status of the patient [2]. Non-differentiated thyroid cancer and medullary thyroid cancer are much more rare and have a particular approach.

## 6. Subclinical Hypothyroidism

### 6.1. Epidemiology/Definition/Diagnostic

Subclinical hypothyroidism (SCH) is defined by the presence of elevated serum TSH levels with normal FT4 or TT4 values. Determining the upper normal limit of TSH in the preconception period and pregnant population remains challenging, but for practical use, many studies use the ATA recommended cut-offs of 2.5–4 mU/L for mild subclinical hypothyroidism and 4–10 mU/L for important SCH for the first trimester [2].

It is a common diagnosis among women of reproductive age, with a prevalence of up to 8%, having an important variability worldwide with geographic region, ethnic origin, body mass index and iodine status [33,39]. Two different tests performed at 6–8 weeks apart, with TSH above the upper normal limit and normal FT4 level, are diagnostic for SCH. The etiology of SCH is similar to that of the overt hypothyroidism, with autoimmune thyroiditis being the main cause, followed by iodine deficiency in iodine deficient areas [31].

### 6.2. Effects on Fertility, Pregnancy and Fetal/Newborn Outcomes

Compared to overt hypothyroidism, which seems to be well associated with an increased risk of infertility and adverse pregnancy and newborn outcomes, the data for SCH are less consistent [31].

Regarding the fertility of women with SCH there is some evidence that suggests a higher prevalence of SCH in infertile women (13.9%) compared with controls (3.9%) [40].

Even though the link between SCH and pregnancy loss or live birth has been previously denied [41], more recent large studies emphasized that pregnant women with SCH were at higher risk for miscarriage, preterm delivery, gestational diabetes, gestational hypertension, eclampsia, and premature rupture of membranes compared with euthyroid women, especially when TSH was >4 mU/L [32,42,43,44,45,46,47]. Moreover, their offspring was more likely to have intrauterine growth restriction, low birth weight, to be admitted in neonatal intensive care unit and have respiratory distress syndrome [43,48].

Regarding the brain development and neurocognitive function of the offspring, existing studies show controversial results. There is proof for an association between higher levels of TSH during pregnancy and a negative impact on neurocognitive function of the offspring [31] and also between maternal SCH and indicators of intellectual disability in offspring [49], but further research is required since other studies showed contradictory results [50,51,52]. Additionally, autoimmunity seems to increase the risk of adverse outcomes [33].

The inconsistency of results between the presence of SCH and adverse outcomes can be partially influenced by the variability of TSH cut-off points used to define SCH, the presence or absence of autoimmunity or the timing of thyroid function evaluation. Additionally, thyroid function may change during pregnancy and as a result, a woman diagnosed with SCH before or at the beginning of her pregnancy may finally progress to overt hypothyroidism or spontaneously revert to euthyroidism [31].

### 6.3. Who and How to Treat?

The decision whether or not to treat women with SCH in the preconception period must be taken with regard to each patient particularities, as well as to existing evidence about the potential benefits and risks over the future pregnancy.

Potential beneficial effects of thyroxine therapy over pregnancy-related morbidity have been suggested over the time in many studies [53,54,55], but TPOAb negative women appeared to have less benefits for the same TSH values [53].

Both 2014 European Thyroid Association Guidelines for the Management of Subclinical Hypothyroidism in Pregnancy and in Children [8] and 2017 Guidelines of the American Thyroid Association for the Diagnosis and Management of Thyroid Disease During Pregnancy and the Postpartum [2] recommend that SCH arising before conception or during gestation should be treated with levothyroxine, especially if TPOAb-positivity associated. In accordance with these recommendations, SCH women with autoimmunity should be treated when TSH > 2.5 mU/L, while SCH women without autoimmunity should be treated when TSH > reference range for non-pregnant before conception or pregnancy-specific reference range after pregnancy confirmed (Figure 2).

Several RCT come to support these recommendations, showing evidence that LT4 treatment in SCH women can significantly reduce the incidence rate of miscarriage [56], decrease the risk of preterm delivery in women who are positive for TPOAb [57] and reduce preterm delivery even in TPOAb negative SCH women with a TSH cut point of 4 mU/L [58]. Regarding the effects of LT4 treatment on child cognition, things still remain unclear, since a recent RCT emphasized no benefit of treatment over the offspring behavior; more than that, children of ‘over-treated’ mothers (FT4 measurement >17.7 pmol/L at either 20 weeks or 30 weeks’ gestation) had significantly more difficulties than untreated [59]. These results suggest that LT4 treatment should be carefully recommended and monitored in order to maximize benefits and minimize potential side effects.

Special attention should be offered to infertile female patients seeking pregnancy and show subtle abnormalities of thyroid dysfunction with/without evidence of thyroid autoimmunity. For those who are attempting natural conception, currently available data support that it is reasonable to consider LT4 treatment for TPOAb negative women with TSH concentrations above the non-pregnant lab reference range, aiming to maintain TSH levels below 2.5 mU/L, given its ability to prevent progression to overt hypothyroidism once pregnancy is achieved. Due to insufficient evidence, the ATA did not make any recommendations for or against levothyroxine therapy in infertile women with TPOAb-positive SCH seeking natural conception [2,60]. On the other hand, in patients undergoing artificial reproductive techniques, in vitro fertilization or intracytoplasmic sperm injection, evidence from RCT emphasize that LT4 therapy improves pregnancy and miscarriage rates in women with SCH, with or without autoimmunity [60,61,62,63,64,65]. Consequently, levothyroxine treatment is strongly recommended in order to maintain a TSH level <2.5 mU/L in these patients [2].

In all cases, levothyroxine treatment should be carefully monitored, with respect to the possible harmful effect of thyroxine overtreatment, which includes preterm delivery, gestational diabetes, hypertension, pre-eclampsia, and ADHD symptoms and behavioral difficulties in children of ‘over-treated’ mother [59,66].

## 7. Euthyroid Autoimmunity

### 7.1. Epidemiology/Definition/Diagnostic

In women of reproductive age, the prevalence of thyroid antibodies is approximately 10–15%, increasing with age [67] and its progression towards overt autoimmune hypothyroidism is a gradual process taking several years [68].

TPOAb are clinically non-functioning IgG antibodies that reflect thyroid autoimmunity and are useful for diagnosing autoimmune hypothyroidism. Pregnant women with thyroid autoimmunity are at risk for developing hypothyroidism because the thyroid’s ability to augment hormone production in pregnancy might be compromised [69,70].

### 7.2. Effects on Fertility, Pregnancy and Fetal/Newborn Outcomes

Thyroid autoimmunity might also have a role in infertility, especially when related to endometriosis or ovarian dysfunction [71], although not all the studies confirmed this association [28,72].

On the other hand, there is increasing data regarding the association of thyroid dysfunction and thyroid autoimmunity with adverse pregnancy outcomes such as miscarriage, recurrent embryo implantation failure and preterm delivery [28,47,71,72,73,74,75,76]. The higher risk of miscarriage could have multiple causes, including a relative thyroid hormone shortage, caused by a decreased thyroid functional capacity, an impaired thyroid response to human chorionic gonadotropin, or a higher general susceptibility to autoimmunity [77,78,79,80]. A dose-dependent relationship between TPOAb and thyroid function as well as the risk of premature delivery has been demonstrated [81]. Additionally, maternal thyroid autoimmunity in early pregnancy might lead to cognitive impairment of the offspring, but further evidence is necessary [82,83,84].

Interestingly, although TPOAb are able to cross the placenta, the maternal passage of either TPOAb or TgAb is not associated with fetal thyroid dysfunction. Since thyroid peroxidase (TPO) is located at the apical surface and in the cytoplasm of thyroid cells, it appears that under normal circumstances, TPOAb do not have access to their autoantigen, which explains the euthyroidism seen in healthy individuals with TPOAb and in neonates born to mothers with high levels of TPOAb. Cell-mediated injury may be necessary for TPOAb to gain access to their antigen and become pathogenic [85]. This is different to what can be seen in some cases of Graves disease, where maternal autoantibodies against the TSH receptor can cross the placenta and interfere with the fetal thyroid gland leading to fetal thyroid dysfunction [86,87,88].

### 7.3. Who and How to Treat?

Whilst the evidence for an association between the presence of TPOAb and increased risks of pregnancy loss is compelling, the results of recent RCT of levothyroxine in euthyroid women with underlying autoimmunity could not demonstrate the beneficial effects of LT4 therapy on live birth rate or recurrent pregnancy losses [72,74,76,86,87,89,90]. Empirical treatment with levothyroxine has been recommended by some researchers, but it has not been universally accepted since the lack of evidence from high-quality clinical trials [91].

On the other hand, numerous clinical papers advocate pro selenium (Se) supplementation during pregnancy in patients with thyroid autoimmunity in order to minimize pregnancy complications such as intrauterine growth restriction, miscarriage, preterm labor, preeclampsia, gestational diabetes, postpartum thyroiditis, and exacerbation of autoimmune thyroid diseases during the first postpartum year, and complications in the fetus (decreased psychomotor development, neural tube defect, cognitive deficits and behavior) [92,93]. Nevertheless, it is important to underline Se’s narrow therapeutic index which may be a problem with regard to the wellbeing of the mother and of the developing fetus [91,92,93].

## 8. Discussion

Thyroid function is essential for a successful conception and a normal fetal development. Current data available is not consistent enough in order to have evidence-based recommendations regarding different thyroid disease preconception or during pregnancy, partly because of ethical considerations and different laboratory cut-off points. Thereafter, further high-quality research through large populational and RCT is necessary about topics such as universal screening vs. case finding for thyroid disease, the rate of progression of SCH to overt hypothyroidism in pregnancy or TSH and TPOAb cut-offs and association with fertility, pregnancy and newborn outcomes. In addition, further evidence is necessary regarding potential benefits of levothyroxine therapy in borderline thyroid abnormalities.

Furthermore, anesthetic issues have to be taken into consideration in patients with overt hypothyroidism: potential compression of the thyroid on the airways, difficult intubation due to the large tongue, increased hypotension due to the anesthetic agents, myocardial depression [94,95]. Among these the hypotension and decreased cardiac output could have the greatest impact on fetus circulation and further brain lesions. Although platelet function is also impaired in hypothyroidism, no side effects have been reported after performing regional anesthesia (spinal and/or epidural anesthesia). Anyway, a preoperative bleeding time should be a standard of care since it is known that acquired von Willebrand disease is the most common coagulation disorder in hypothyroidism [96,97]. When performing regional anesthetic techniques, special attention should be on the hypotension occurrence such as vasopressor (phenylephrine or ephedrine) and fluids should be prepared. If the pregnant woman has contraindications for regional anesthesia, general anesthesia is the other option. In this situation (especially in scheduled surgery or Cesarean-section) an ear, nose and throat (ENT) consultant should be requested for larynx and vocal cords assessment, less sedation should be given preoperative due to potentially respiratory depression, premedication with H_2_ antagonists and metoclopramide is recommended because of slow gastric emptying. Special attention should be paid to hypotension occurrence following anesthetic induction by a combined mechanism of an impaired baroreceptor function and myocardial depression so a preoperative assessment of the volemic status (using noninvasive technique as echocardiography) is mandatory and vasopressors and fluids should be available. Regarding the mandatory invasive blood pressure monitoring, there is no consensus in these cases so a three-minute noninvasive blood pressure (NIBP) monitoring test is acceptable. In order to prevent the over dosage of the anesthetic drugs, is the anesthetic depth monitoring (e.g., BIS, entropy) during the intervention [98]. Another issue is represented by hypothermia, so warm blankets must be available in the operation room.

## 9. Conclusions

Current recommendations according to the most recent data available about preconceptional counseling in women with thyroid disease can be synthetized as follows:-Screening for thyroid disease is recommended only in the presence of risk factors;-In most regions, women should supplement their diet with a daily oral supplement of potassium iodide that contains 150 µg of iodine, or even 250 µg in areas with severe iodine deficiency;-Hypothyroid women receiving LT4 treatment should adjust doses in order to have TSH values between the lower reference limit and 2.5 mU/L and also be informed that in case of suspected or confirmed pregnancy they should independently increase the LT4 dose by 20–30% (e.g., by doubling dose on 2 days a week);-Differentiated thyroid cancer should not discourage intended pregnancy nor indicate abortion. Conception should, however, occur after remission is confirmed and after adequate thyroid hormone replacement has been attained;-TPOAb positive SCH women should be treated when TSH > 2.5 mU/L, while TPOAb negative SCH women should be treated when TSH > reference range for non-pregnant before conception;-For TPOAb negative SCH women LT4 treatment could be considered;-Patients undergoing artificial reproductive techniques should be treated with LT4 in order to maintain a TSH level <2.5 mU/L;-In euthyroid women with positive TPOAb, empirical treatment with levothyroxine has been recommended by some researchers, but it has not been universally accepted due to lack of high-quality evidence from RCT and potential side-effects on the offspring.

## Figures and Tables

**Figure 1 medicina-58-01122-f001:**
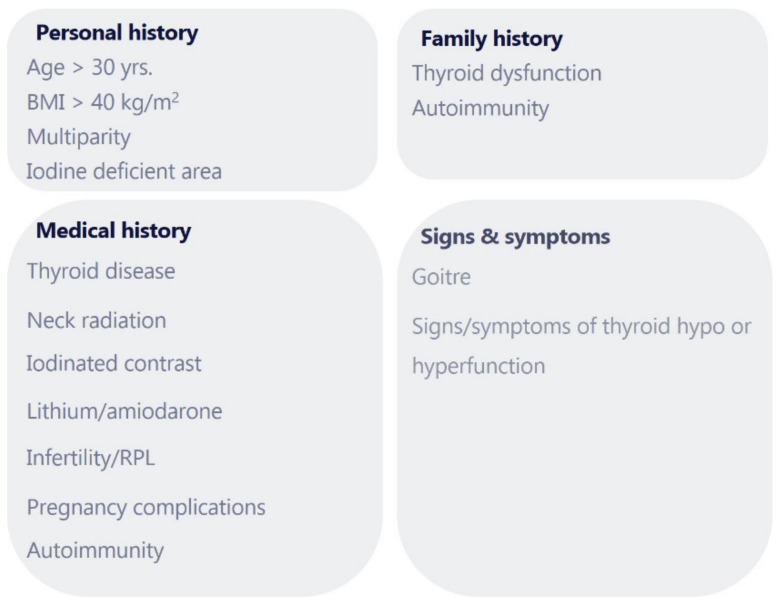
Risk factors for thyroid disease in the preconception period; Yrs—years; BMI—body mass index; RPL—recurrent pregnancy loss.

**Figure 2 medicina-58-01122-f002:**
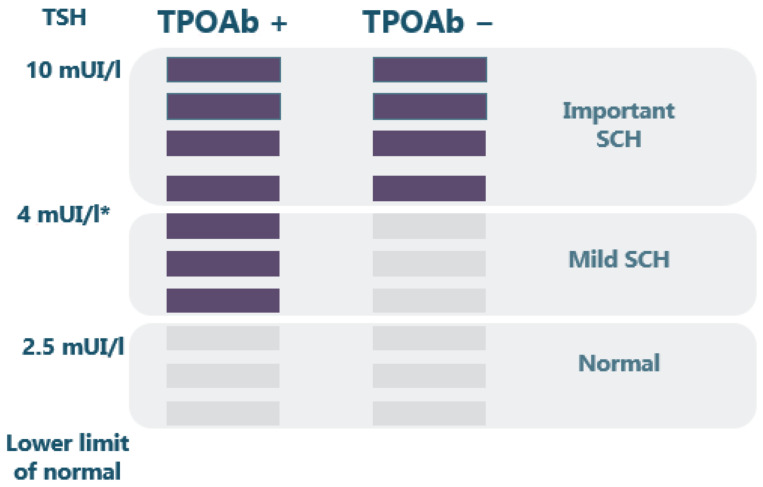
SCH approach according to TSH value and the presence of thyroid autoimmunity—filled violet rectangles suggest treatment while grey rectangles suggest no need for treatment; SCH—subclinical hypothyroidism; TSH—thyroid stimulating hormone; TPOAb—thyroid peroxidase antibodies; * —or upper limit of normal for nonpregnant.

## Data Availability

Not applicable.

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
