# Peer review of "Preconception Counseling in Patients with Hypothyroidism and/or Thyroid Autoimmunity"

_medicina, 2022, doi:10.3390/medicina58081122_

Round 1

Reviewer 1 Report

This is a well written manuscript presenting interesting and valuable information. However, I have a few suggestions for the authors. 

1. When elaborating on endemic iodine deficiency, the authors only mention Carpathean regions. What about other parts of Europe/world?

2. What are the recommendations for thyroid tumor patients who are about to go through (or already went through) thyroidectomy?

Reviewer 2 Report

The manuscript highlights the importance of preconception counselling and screening of women in reproductive age with thyroid dysfunction,  for preventing complications during pregnancy and the adverse outcomes. The authors have done a great job compiling the available literature to support their study. 

In order to reach larger audience, authors are suggested to explain little more about the impact of their study and how it is adding to the available literature.  
